# Vitamin D Content in Commonly Consumed Mushrooms in Thailand and Its True Retention after Household Cooking

**DOI:** 10.3390/foods12112141

**Published:** 2023-05-25

**Authors:** Piyanut Sridonpai, Philipda Suthipibul, Konpong Boonyingsathit, Chanika Chimkerd, Sitima Jittinandana, Kunchit Judprasong

**Affiliations:** 1Institute of Nutrition, Mahidol University, Salaya, Phutthamonthon, Nakhon Pathom 73170, Thailand; piyanut.sri@mahidol.ac.th (P.S.); s.philipda@gmail.com (P.S.); konpong.boo@mahidol.ac.th (K.B.); sjittinandana@gmail.com (S.J.); 2Center of Analysis for Product Quality, Natural Products Division, Faculty of Pharmacy, Mahidol University, Sri-Ayuthaya Road, Rajathevi, Bangkok 10400, Thailand; chanika.chm@mahidol.ac.th

**Keywords:** vitamin D, ergosterol, mushroom, cooking method, true retention

## Abstract

This study investigated the vitamin D level of nine species of cultivated mushrooms and three species of wild mushrooms commonly consumed in Thailand and the effect of cooking on their vitamin D content. Cultivated mushrooms were obtained from three wholesale markets, while wild mushrooms were collected from three trails in a conservation area. Mushrooms from each source were separated into four groups: raw, boiled, stir-fried, and grilled. Different forms of vitamin D were analyzed using liquid chromatography with tandem mass spectrometry (LC-MS/MS). The analyzed method demonstrated good linearity, accuracy, and precision, as well as being low in the limit of detection and limit of quantitation. Results showed that vitamin D2 and ergosterol (provitamin D2) were the major forms of vitamin D found in the mushrooms. Both raw cultivated and wild mushrooms had wide ranging ergosterol contents (7713–17,273 μg/100 g edible portion, EP). Lung oyster mushroom and termite mushroom contained high levels of vitamin D2 (15.88 ± 7.31 and 7.15 ± 0.67 μg/100 g EP, respectively), while other mushroom species had negligible amounts (0.06 to 2.31 μg per 100 g EP). True retention (TR) levels of vitamin D2 after boiling, stir-frying, and grilling were not significantly different (*p* > 0.05) (with estimated marginal means ± standard error 64.0 ± 2.3%, 58.8 ± 2.3%, and 64.7 ± 3.6% TR, respectively). Consuming cooked lung oyster mushrooms, in particular, along with regular exposure to sunlight should be promoted to reduce the incidence of vitamin D deficiency.

## 1. Introduction

Vitamin D comprises a group of fat-soluble vitamins that take several forms. The two major forms are vitamin D2 (ergocalciferol) which is found in certain fungi including wild mushrooms, and vitamin D3 (cholecalciferol), which is found mainly in animal-based foodstuffs (e.g., fish, shrimp, and cod liver oil) [1]. Vitamin D is categorized as a vitamin that can be produced naturally in humans from 7-dehydrocholesterol to pro-vitamin D3 when the skin is exposed to ultraviolet (UV) radiation from sunlight and provided in sufficient quantities to maintain adequate vitamin D status (usually between 50% and 90%) [2]. For foodstuffs, Sridonpai P. et al. [3] reported that Nile tilapia, common silver barb, and Red Nile tilapia contain extremely high levels of vitamin D3 (19.8, 31.0, and 48.5 μg per 100 g edible portion, respectively). Vitamin D3 is absorbed in the liver to 25-hydroxyvitamin D (25(OH)D), which is the major metabolized form of vitamin D and is used for determining an individual’s vitamin D status [4]. However, if sunlight exposure is limited in terms of cutaneous vitamin D production, for example through aging, skin pigmentation, time of day, sunscreen usage, season, or latitude, food sources of vitamin D are essential to maintain healthy circulating 25(OH)D concentrations.

Vitamin D deficiency is a worldwide public health problem and adequate dietary intake has gradually become important. It is estimated that one billion people globally are deficient in vitamin D (25(OH)D concentrations less than 50 nmol/L) with a prevalence of up to or over 50% being reported in some population-based studies [5]. For example, studies have reported vitamin D deficiency to be in approximately 25% of the population in Canada, 31% in Australia, 45–52% in New Zealand, 22–36% in the USA, and 47–65% in Republic of Korea [6]. A large-scale assessment of vitamin D status in Thailand reported a incidence rate of 45.2% for vitamin D insufficiency (25(OH)D less than 75 nmol/L) and 5.7% for vitamin D deficiency (25(OH)D less than 50 nmol/L) [7].

Currently, a knowledge gap exists in Thailand’s food composition database in terms of mushrooms and their vitamin D content. Closing this gap is important, since many studies have reported mushrooms as good dietary sources of vitamin D, especially vitamin D2 [8,9]. In general, all mushroom species contain ergosterol (provitamin D2) but the amount varies widely. For example, Phillips KM et al. [10] reported that edible varieties of mushrooms from retail shops sold in the UK and especially cultivated mushrooms (e.g., portabella (*Agaricus bisporus*) and common white-button) grown in atmospherically controlled growing rooms and darkness did not change in terms of vitamin D2 levels after exposure to sunlight or UV radiation. Further, the vitamin D2 content of fresh white button mushrooms was reported to be less than 20 IU (0.5 μg per 100 g). In contrast, huge amounts of vitamin D2 have been found in wild mushrooms including *Cantharellus cibarius*, at 10.7 μg per 100 g fresh weight (FW), chanterelles, at 21.1 μg per 100 g FW, and *Boletus edulis*, at 58.7 μg per 100 g FW [11]. Edible mushrooms are a good source of vitamin D and other nutrients, which can be linked to Sustainable Development Goal (SDG) number 3: good health and well-being. In addition, mushrooms are also involved in SDG 2: zero hunger due to the fact that mushrooms are easy to cultivate, promote consumption, and provide an opportunity to diversify diets, particularly for communities with limited access to animal-based proteins.

To our knowledge, however, no information exists on the vitamin D and ergosterol content in mushrooms commonly consumed in Thailand. Consequently, this study firstly aimed to assess vitamin D content in different varieties of commonly consumed mushrooms in Thailand to identify those types that are good sources of vitamin D and can be promoted for consumption to potentially enhance vitamin D status. Moreover, the Thai people do not commonly consume uncooked mushrooms, but prepare them by boiling, stir-frying, or grilling (Food Consumption Data of Thailand [12]). Consequently, this study also assessed the effect of these cooking methods on vitamin D content.

## 2. Materials and Methods

### 2.1. Chemicals

All organic solvents and reagents used throughout this study for sample and mobile phase preparation were analytical grade. The organic solvents were pentane (RCI LABSCAN, HPLC grade), ethyl ether (LOBA CHEMIE), acetonitrile (HPLC grade), methanol (both of LC-MS and HPLC grade), and sodium L-ascorbate (crystalline > 98%, Sigma-Aldrich Corp., St. Louis, MO, USA). The reagents were potassium hydroxide (45%) and sodium chloride (10%, Ajax Finechem, Taren Point, Australia).

Standards were purchased at the highest purity (>95%) from suppliers, including vitamin D3 (product No. 47763), vitamin D2 (Suplelco, Bellefonte, PA, USA, product No. 47768), tri-deuterated vitamin D2 used as an internal standard (2H_3_-D_2_) (Sigma-Aldrich Corp., St. Louis, MO, USA, product No. V-026-1ML), ergosterol (ThermoFisher, Winsford, UK, product No. B23840), 7-DHC (Sigma-Aldrich-Corp., St. Louis, MO, USA, product No. 30800), 25-OH D2 (Enzo, New York, NY, USA, product No. BML-DM101), and 25-OH D3 (Enzo., New York, NY, USA, product No. BML-DM100).

### 2.2. Selection and Collection of Cultivated and Wild Mushrooms

Nine species of the most commonly consumed cultivated mushrooms and three species of wild mushrooms (Figure 1) were selected based on combined data from the most commonly consumed in the Food Consumption Data of Thailand [12] and lack of database in the Food Composition Database of Thailand [13]. These mushroom varieties are shown in Table 1. Simple random sampling and sampling methods from a previous report [14] were used as advice for the sample collection. The 12 species of selected mushrooms, at 4–5 kg each, were obtained from July to December 2019. The cultivated species were purchased from three main markets and three sellers (*n* = 3) covering the Thai Market (representative of east and/or northern parts of Thailand), Klong Toey Market (representative of the Bangkok area), and Railway Bangkok-Noi Market or Sala Nam Ron Market (representative of southern parts of Thailand). Wild mushrooms were collected from three trails (*n* = 3) in the conservation area of Srinagarind Dam located in Kanchanaburi Province, Thailand.

### 2.3. Cooking Processes

The mushrooms were prepared using common household practices, i.e., trimming, washing once with tap water, and then washing twice with drinking water. Each mushroom species was prepared separately. Edible portions were collected, weight recorded, and equally divided into 4 samples. One sample was prepared as a raw sample where the edible part was cut into small pieces, then mixed in a food blender (Wongdec, WTI-1684A, Nonthaburi, Thailand), put in a screw-cap plastic bottle, and stored at −20 °C until analysis of moisture. The other 3 samples were prepared and cooked by boiling, stir-frying, and grilling according to common household methods (Table 2).

For boiling, an aluminum pot (28 cm diameter) was filled with 2–3 L of deionized water, which was boiled. After boiling, the prepared mushrooms were put into the pot and left until they were cooked. The samples were then drained for 30 s and left for 10 min before weighing.

An iron wok (26 cm diameter) was used for stir-frying. Whole mushroom samples were cut into small sizes and then added to 30 milliliters of heated soybean oil (for 500 g of fresh mushrooms per batch). The mushrooms were then stirred until completely cooked.

An open electric grill (TEBA, GRIGLIA 2000, Boreal, Singapore) was used for grilling. After the grill temperature was stable, small pieces of mushroom were placed on the top rack of the grill and grilled until completely cooked.

All mushrooms were cooked without the addition of any other ingredients or seasonings. The temperature in the middle of the mushrooms and heating temperature were measured using a digital thermometer (OAKTON, TEMP 10J, Cole-Parmer, Burlington Township, NJ, USA). Cooking methods and conditions for each mushroom variety are shown in Table 2.

The raw and cooked mushrooms were then homogenized in a food blender. For freeze drying samples, homogenized samples were placed on freeze-dryer trays and stored in a freezer at −80 °C overnight prior to transfer to the freeze dryer operating system. Freeze-dried mushroom samples were dried in the freeze dryer system (Heto Powerdry PL 9000 Freeze Dryer, Corston, UK) for 2–3 days or until completely dried. The weight of each sample before and after freeze drying was recorded. Each mushroom sample was re-homogenized in a food blender, filled in aluminum foil bags, and stored at −20 °C until analysis of vitamin D.

### 2.4. Determination of Vitamin D by LC-MS/MS

An ultimate 3000 UHPLC system (ThermoFisher, MA, USA), equipped with a Water HSST 3 reversed-phase column 2.1 nm × 150 nm, 1.7 μm particle size, was used for separating the different forms of vitamin D. The mobile phase was composed of 2 mM ammonium formate in water (mobile phase A) and 2 mM ammonium formate in methanol (mobile phase B). The flow rate of the gradient elution was 500 μL/min, the temperature of the column was set at 60 °C, and the volume for injection was 20 μL. The gradient system was started with 10% A, 90% B held for 3 min, changed to 0% A, 100% B for 8 min, and changed to 10% A, 90% B for 12 min of running time.

A TSQ Quantis ion trap tandem mass spectrometer (Thermo Fisher, Waltham, MA, USA) was used for the quantitative measurement of vitamin D. Atmospheric pressure chemical ionization (APCI) was applied as the ionization method in mass spectrometry. Quantitative analysis was measured by the MS/MS mode for all forms of vitamin D. The following instrument parameters were used: vaporizer temperature (350 °C), iron transfer tube temperature (325 °C), auxiliary gas (5 Arb), sheath gas (45 arbitrary units, Arb), sweep gas (1 Arb), positive ion discharge current (6.5 μA), and negative ion discharge current (10 μA). The parent and daughter ions of vitamin D2, vitamin D3, ergosterol, 7-DHC, 25-OH D2 25-OH D3, and internal standard, as well as collision energy are shown in Table 3.

### 2.5. Quality Control System of Vitamin D Analysis

All samples were analyzed, in duplicate, at the accredited ISO/IEC 17025:2017 laboratory of the Institute of Nutrition, Mahidol University. Method validations covered the limit of detection (LOD), limit of quantitation (LOQ), linearity range, accuracy, precision, and percent recovery. Standard reference material (SRM) (NIST-1846 infant formula, MD, USA) was used as a certified reference material for accuracy determination and as a quality control (QC) sample for every set of analyses. Each mushroom sample was analyzed in duplicate along with the QC sample (SRM). The vitamin D value of the QC sample for each analytical batch had to be within the mean ± 2 standard deviations (SD) of the certified value. If the vitamin D value fell outside the mean ± 2SD, the analysis of the unknown analyzed sample was repeated.

### 2.6. True Retention of Vitamin D in Mushrooms

Samples were weighed before and after cooking (to at least 4 significant digits). The data achieved, calculated with the amounts of vitamin D in raw and cooked mushrooms, were used to compute true retention [15] as per the formula below.
$$\%\;\mathrm{True}\;\mathrm{retention}\;=\frac{\mu g\;\mathrm{vitamin}\;D\;\mathrm{per}\;100\;g\;\mathrm{of}\;\mathrm{cooked}\;\mathrm{mushroom}\;\times \;\mathrm{weight}\;\mathrm{of}\;\mathrm{cooked}\;\mathrm{mushroom}}{\mu g\;\mathrm{vitamin}\;D\;\mathrm{per}\;100\;g\;\mathrm{of}\;\mathrm{raw}\;\mathrm{mushroom}\;\times \;\mathrm{weight}\;\mathrm{of}\;\mathrm{raw}\;\mathrm{mushroom}}\times 100$$

### 2.7. Statistical Analysis

All data for yield factors, edible portions, moisture, and vitamin D content of all mushrooms from 3 markets were presented as mean ± standard deviation (SD). Different types of mushrooms and their true retention after cooking were tested using two-way ANOVA with interaction followed by Tukey’s Honestly Significant Difference to test multiple pairwise comparisons. The statistical analysis was tested using IBM^®^ (Hampshire, UK) SPSS Statistics for Windows, Version 21.0.

## 3. Results and Discussion

### 3.1. Performance Characteristics of the Analytical Method for Vitamin D by LC-MS/MS

Calibration curves for all forms of vitamin D showed good linearity (R^2^ or coefficient of determination values greater than 0.995). Repeatability and intermediate precision (different days) were obtained with a relative standard deviation of less than seven and 13%, respectively. Limits of detection (LOD) of 25-OH D3, 25-OH D2, 7-dehydrocholesterol, ergosterol, vitamin D3, and vitamin D2 were 0.09, 0.14, 0.12, 0.36, 0.05, and 0.06 μg/100 g, respectively. The limits of quantitation (LOQ) were 0.29, 0.47, 0.40, 1.19, 0.18, and 0.20 μg/100 g, respectively. The accuracy performance of this method was assessed using standard reference material (NIST-1846, infant formula). The result for vitamin D showed no significant difference (*p* > 0.05) from that of the certified value. Vitamin D3 content analyzed by LC-MS/MS in this study was 0.117 ± 0.01 mg/kg, which was not significantly different from that of the certified value (0.111 ± 0.01 mg/kg vitamin D3) (*p* = 0.480). Consequently, this method is suitable for determining different forms of vitamin D in the samples.

### 3.2. Edible Portion

The edible portion of each mushroom variety is the part used in normal household cooking as collected or as purchased, expressed on a wet weight basis. For the morphology of mushrooms, most of the edible portion comprises the partial veil including part of the stalk, cap, and gills that have high spore production and ergosterol. Table 4 gives the percent edible portions of raw and cooked cultivated and wild mushrooms, which varied depending on the types and characteristics of mushroom species. Nonetheless, percent edible portions of the cultivated and wild mushroom species were within the same range (more than 80–100%). For moisture and vitamin D analyses, therefore, a whole mushroom was prepared as an edible portion.

### 3.3. Yield Factor

Accurate recipe calculation procedures are achieved by the use of yield factors for inputting nutrient values for composite foods. Boiled, stir-fried, and grilled mushrooms retained high yield factors (0.61–1.07, 0.79–0.94, and 0.61–0.88, respectively) (Table 4). For both cultivated and wild mushrooms, grilling retained a lower yield factor compared with the other cooking methods. The cooking yield here was similar to that reported in the Ložnjak and Jakobsen [16] study, which investigated the stability of vitamin D2 in mushrooms after cooking. They reported low yield factors for boiling, pan-frying, and baking bio-fortified mushrooms (0.74–0.76, 0.65–0.73, and 0.51–0.52, respectively). Consequently, different yield factors could be affected by different cooking methods for both wild and cultivated mushrooms.

### 3.4. Moisture Content

The moisture contents of cultivated and wild mushrooms were 89–90 and 82–94 g per 100 g edible portion (EP), respectively (Table 4). Similar moisture content levels were reported in the United States Department of Agriculture Food Composition Database 2021 [9] (88.6, 90.4, 89.2, and 91.8 g per 100 g EP for raw shiitake, maitake, oyster, and white button mushrooms, respectively). Boiling for both cultivated and wild mushrooms led to slightly higher moisture contents (ranging between 86 and 91 g per 100 g EP). Stir-frying and grilling for all mushrooms, however, led to a major loss of water that resulted in lower moisture contents compared with boiling (83–85 and 86–88 g per 100 g EP, respectively).

### 3.5. Ergosterol Content

The amounts of ergosterol (precursor vitamin D2) in raw and cooked cultivated and wild mushrooms per 100 g EP are shown in Table 5. A high amount of ergosterol was found in both wild and cultivated mushrooms. The ergosterol content in raw cultivated mushrooms ranged from 8480 to 12,973 μg per 100 g EP. In raw wild mushrooms, it ranged from 7713 to 17,273 μg per 100 g EP. Notably, the amount of ergosterol in this study was not directly related to the amount of vitamin D2. Although the raw hygroscopic earthstar mushroom contained the highest amount of ergosterol (17,273 ± 662 μg per 100 g EP), the amount of vitamin D2 was generally low (0.07 ± 0.01 μg per 100 g EP). A similar observation was found for raw Bhutan, enokitake, and shiitake mushrooms (0.07 ± 0.01, 0.06 ± 0.01, and 0.17 ± 0.08 μg of vitamin D2 per 100 g EP, respectively). Jasinghe and Perera [17] reported that, in raw shiitake mushrooms, ergosterol concentrations were highest in the gills, stalk, and cap, which can generate a high amount of vitamin D2 (22.8 μg per g dried matter) when exposed to UV-B radiation. Consequently, variation in vitamin D2 content in mushrooms is generally dependent on such factors as time of day, latitude, season, exposure time, weather conditions, surface area, and the morphology of each type of mushroom.

### 3.6. Vitamin D Content

The amounts of vitamin D2 in raw and cooked cultivated and wild mushrooms per 100 g EP are shown in Table 5. Commercially cultivated mushrooms are typically grown in darkness. They are likely to be exposed to light during harvesting but usually receive little to no UV radiation. After harvesting, these raw mushrooms are usually transported to markets and retail outlets under refrigerated transport. In Thailand, some mushroom varieties (e.g., straw, oyster, shiitake, and log white fungi mushrooms) are cultivated in controlled growing rooms using different mushroom culture recipes and require a dark environment to grow. Some mushrooms, e.g., button and shiitake mushrooms, require darkness during mushroom production [18], which preserves the moisture that mushroom spores need to reproduce.

Mushrooms are the only non-animal foods that can make vitamin D. Moreover, they contain a provitamin or ergosterol (precursor) that is converted into vitamin D when exposed to the sun’s UV radiation, like human skin that synthesizes vitamin D upon sun exposure. High levels of provitamin D2 or ergosterol have been found in abalone, lung oyster, termite, and hygroscopic earthstar mushrooms.

In this study, most varieties of mushrooms had negligible amounts of vitamin D2, ranging from 0.06 to 2.31 μg per 100 g EP, except for lung oyster mushrooms, which contained an extraordinarily high level of vitamin D2 (15.88 ± 7.31 μg per 100 g EP). This wide variation in vitamin D2 content can be related to cultivation practices, precursor (ergosterol) content of vitamin D2, and the physical characteristics of each mushroom variety. In Hungary, Gyôrfi [19] reported increasing the vitamin D2 level of oyster mushrooms using UV light and found vitamin D2 in controlled raw oyster mushrooms (non-UVB treatments) at a similar value (10 to 20 μg per 100 g FW). Most vitamin D2 content of cultivated mushrooms in this study agreed well with those reported in studies of raw cultivated mushrooms collected from retail shops sold in the UK, North America, Europe, New Zealand, and Australia, which commonly reported less than 1 μg per 100 g FW [11,20,21,22,23]. Moreover, the National Nutrient Database of the United States (Department of Agriculture 2021) [9] also reported low amounts of vitamin D2 in cultivated mushrooms (i.e., white button, oyster, shiitake, and maitake mushrooms containing vitamin D levels of 0.02, 0.04, 0.06, and 1.6 μg per 100 g FW, respectively). Banlangsawan & Sanoamuang [24] studied the effect of UV-B irradiation in Thai edible mushrooms, namely enokitake mushroom (*Flammulina velutipes*), log black fungi mushroom (*Lentinus polychrous* Lev), log white fungi mushroom (*Lentinus squarrosulus* Mont), wood ear mushroom (*Auricularia auricula-judae*), abalone mushroom (*Pleurotus ostreatus* (FR.)), and lung oyster mushroom (*Pleurotus pulmonarius*), and these raw mushrooms had vitamin D2 levels of less than 4 μg/g of dry weight. Therefore, the vitamin D contents found in all mushrooms in this study agree well with previous studies [9,11,20,21,22,23,24]. However, cultivated mushrooms are often available in markets, whereas some species of wild mushrooms are grown and harvested during a specific season and in a specific geographic area and should be accurately identified in order to avoid poisonous varieties.

Wild mushrooms in this study contained vitamin D2 content ranging from 0.07 ± 0.01 μg per 100 g EP in hygroscopic earthstar mushrooms to 7.15 ± 0.67 μg per 100 g EP in termite mushrooms (Table 5). Hygroscopic earthstar mushrooms have a circular shape with a hard thick shell (Figure 1), which may hinder sunlight penetration and cause low vitamin D2 content. Most wild mushrooms usually grow on decomposing leaves, trees, dung, soil, mulch, or compost with appropriate temperature, light, water, and humidity. The growing conditions for wild half-dyed slender Caesar and termite mushrooms allow more opportunity to receive sunlight or UV radiation, leading to high natural vitamin D2 content. Termite mushrooms (wild mushrooms) were found to contain a high variation of ergosterol and vitamin D2 (13,044 ± 3004 and 7.15 ± 0.67 μg/100 g EP, about 23% and 9% RSD, respectively). These are underground mushrooms and are typically located near the base of trees and termite areas. They were collected in different areas of the conservatively studied area. The intensity of sunlight and UV radiation that the mushrooms are exposed to during growth can impact the high variation of production of vitamin D. The levels of vitamin D in termite mushrooms can also vary depending on the season; these were collected during the summer season. Overall, the levels of vitamin D in termite mushrooms can vary due to various factors, including geographic location, growing conditions, and seasonal variability. Since this study is the first to report the ergosterol and vitamin D content in wild Thai mushrooms, no comparative data from other studies exist for these same mushrooms. However, huge amounts of vitamin D2 have been reported in *Boletus edulis* (58.7 μg per 100 g FW) and wild funnel chanterelles (21.1 μg per 100 g FW) [11].

Lung oyster mushrooms and termite mushrooms contained high levels of vitamin D2 (15.88 ± 7.31 and 7.15 ± 0.67 μg per 100 g EP, respectively). Lavelli et al. [25] summarized in vivo studies on the effects of vitamin-D-fortified foods. There is a limit to using vitamin D2 for in vivo studies. Vitamin D2 from UV-irradiated yeast in bread (25 μg/d) did not raise serum 25(OH)D concentration [26]. Biancuzzo et al. [27] indicated the equal bioavailability of vitamin D2 and vitamin D3 in Ca^++^-enriched orange juice and capsules. Vitamin D3 supplement doses were shown to have an effect in increasing serum 25(OH)D concentration, for example 10 μg/d in Ca^++^-enriched yogurt studied in healthy women aged >65 years [28], 10.8 μg/d in low-fiber wheat bread and 12.3 μg/d in high-fiber sourdough rye bread studied in healthy women aged 25–45 years [29], and 15 μg/d in cheese studied in healthy women aged 25–45 years [30]. Therefore, natural vitamin D in these mushrooms are in the range of the fortified functional foods.

### 3.7. Effect of Different Cooking Procedure on Vitamin D in Mushrooms

The true retention of vitamin D2 in mushrooms after cooking is the proportion of vitamin D2 remaining compared with the amount originally present in a given weight of the mushrooms before cooking. It can increase or decrease from 100 percent depending on the characteristics of each mushroom and vitamin D2 content. Results for the true retention of vitamin D2 due to cooking are shown in Table 5. Lung oyster mushrooms were the only mushrooms that contained an extraordinarily high level of vitamin D2. However, the true retention of vitamin D2 in boiled, stir-fried, and grilled lung oyster mushrooms (64.3 ± 30.9%, 27.0 ± 5.6%, and 59.5 ± 36.0% TR, respectively) was low after heat processing. Ložnjak and Jakobsen [16] also investigated the true retention of vitamin D2 in bio-fortified mushrooms (*Agaricus bisporus*), entailing boiling at different pH levels, steam, pan-frying, cooking, and oven baking. In the bio-fortified mushrooms, the true retention of vitamin D2 was reported to be 62–88%. Jakobsen and Knuthsen [31] studied true retention in vitamin D2-enriched bread after heating in an oven, which showed a high retention (73–89% TR).

Based on physical characteristics, straw, log white fungi, and termite mushrooms contain soft parts that could lead to the loss of vitamin D2 during cooking. Hence, true retention due to boiling (4.3–16.2% TR) and stir-frying (4.4–9.4% TR) in these mushrooms was much lower than for the other mushrooms due to their soft structure that is sensitive to heat exposure during cooking. Consequently, vitamin D2 in both cultivated and wild mushrooms significantly decreased after cooking. Loss of vitamin D2 using household cooking practices, therefore, can be due to high cooking temperatures as well as leaching into cooking oil during stir-frying.

For Bhutan oyster mushrooms, vitamin D2 remarkably increased after boiling, stir-frying, and grilling (15, 17, and 39 times compared with the raw, respectively) whereas ergosterol increased only 2–3 times. This may be because when certain foods that contain vitamin D are cooked, the heat can cause some of the vitamin D that is present to become more available and easier to extract and detect. Another reason is that when mushrooms are exposed to heat they may convert a form of vitamin D called ergosterol into a more usable form of vitamin D2.

After determining differences in vitamin D content for all mushrooms by two-way ANOVA with interaction followed by Turkey’s HSD post hoc test, a significant difference in the combined effects of mushroom species and cooking methods (Table 6) was found (*p* = 0.001). A significant difference in vitamin D content was evident among different types of mushrooms (*p* < 0.05) (Table 6). Lung oyster mushrooms had the highest significant vitamin D level (*p* < 0.05, estimated marginal means was 11.45 μg per 100 g EP) compared with other mushrooms (0.06–3.01 μg per 100 g EP). There was also a significant difference in the effect of cooking methods on vitamin D content (*p* < 0.001) (Table 6, estimated marginal means ranged from 0.87 to 3.13 μg per 100 g EP). Stir-fried mushrooms had the lowest significant vitamin D levels (*p* < 0.05, estimated marginal means was 0.87 μg per 100 g EP), which may be due to their structure that promotes loss of vitamin D when exposed to heat during cooking. Vitamin D content of mushrooms can vary widely depending on several factors, including the type of mushroom, the growing conditions, cooking methods and processing, and storage methods used.

## 4. Conclusions

This study assessed vitamin D content and the effects of different cooking methods on true retention for selected commonly consumed mushrooms in Thailand. Cultivated mushrooms grown and harvested in darkness and controlled rooms contained low levels of vitamin D2, except for the lung oyster mushroom and termite mushroom, which had extraordinarily high levels. The wild mushroom varieties also contained low levels of vitamin D2. Ergosterol content in the cultivated and wild mushrooms varied widely depending on the variety of mushroom species. True retention levels of vitamin D2 in boiled, stir-fried, and grilled mushrooms were not significantly different. Cooking methods may cause a significant loss of vitamin D2, but the degree of loss depends on the physical characteristics of a mushroom, the heating process, and cooking time. Cooked lung oyster mushroom is the most suitable for promotion as a high source of vitamin D, since it is easy to consume, has a delicious taste, and is inexpensive. It should be promoted especially for persons who have limited exposure to the sun and thus require more vitamin-D-rich foods.

## Figures and Tables

**Figure 1 foods-12-02141-f001:**
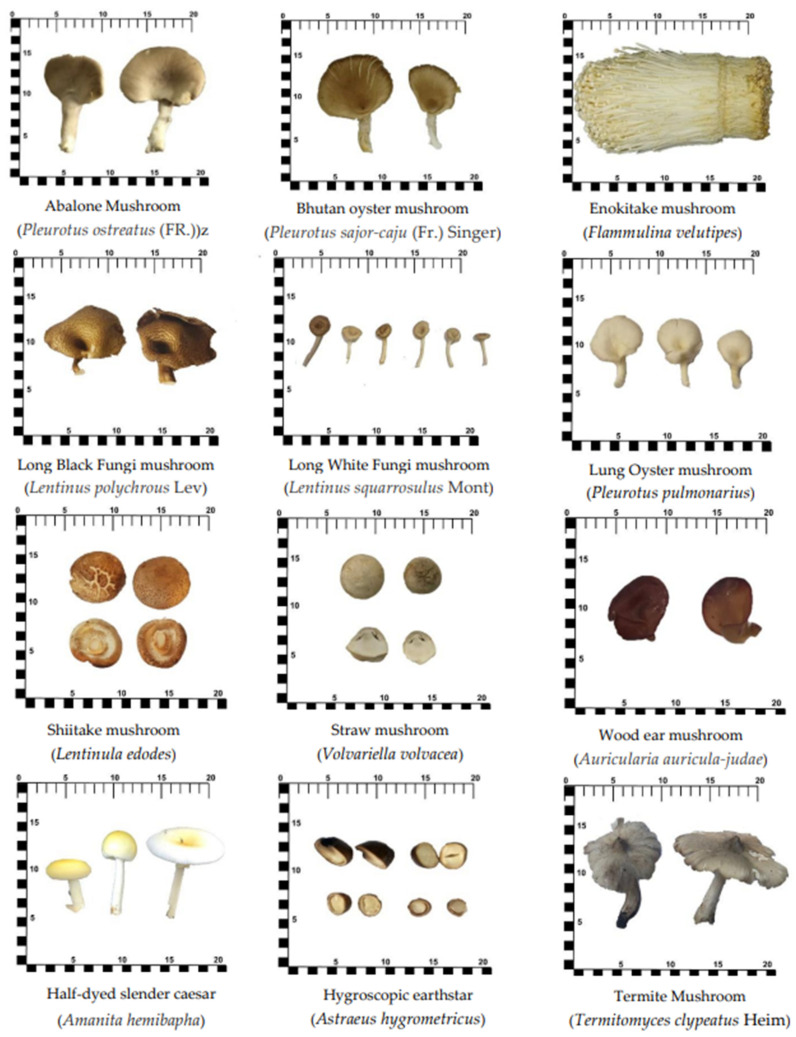
Pictures of all studied mushrooms.

**Table 1 foods-12-02141-t001:** List of selected cultivated and wild mushrooms.

Mushroom Type	English Name	Thai Name	Scientific Name
Cultivated mushrooms	Abalone mushroom	Het-Pao-Heu	*Pleurotus ostreatus* (FR.)
	Bhutan oyster mushroom	Het-Bhutan	*Pleurotus sajor-caju* (Fr.) Singer
	Enokitake mushroom	Het-Khem-Thong	*Flammulina velutipes*
	Log black fungi mushroom	Het-Kra-Ddang	*Lentinus polychrous* Lev
	Log white fungi mushroom	Het-Khon-Khao	*Lentinus squarrosulus* Mont
	Lung oyster mushroom	Het-Nang-Fa	*Pleurotus pulmonarius*
	Shiitake mushroom	Het-Hom	*Lentinula edodes*
	Straw mushroom	Het-Fang	*Volvariella volvacea*
	Wood ear mushroom	Het-Hu-Nu	*Auricularia auricula-judae*
Wild mushrooms	Half-dyed slender Caesar	Het-Ra-Ngok	*Amanita hemibapha*
	Hygroscopic earthstar	Het-Pho	*Astraeus hygrometricus*
	Termite mushroom	Het-Khon	*Termitomyces clypeatus* Heim

**Table 2 foods-12-02141-t002:** Common household methods and conditions for cooking mushrooms.

Mushroom	Cooking Method, Time (Min.), and Temperature		
	Boiling (85–100 °C)	Stir-Frying (110–150 °C)	Grilling (150–200 °C)
Cultivated mushrooms:			
Abalone mushrooms (*Pleurotus ostreatus* (FR.))	6 ^a^	5	- ^b^
Bhutan oyster mushroom (*Pleurotus sajor-caju* (Fr.) Singer)	10	4	6
Enokitake mushroom (*Flammulina velutipes*)	4	3	5
Log black fungi mushroom (*Lentinus polychrous* Lev)	10	4	-
Log white fungi mushroom (*Lentinus squarrosulus* Mont)	6	3	
Lung oyster mushroom (*Pleurotus pulmonarius*)	10	4	6
Shiitake mushroom (*Lentinula edodes*)	8	4	10
Straw mushroom (*Volvariella volvacea*)	10	5	8
Wood ear mushroom (*Auricularia auricula-judae*)	5	3	-
Wild mushrooms:			
Half-dyed slender Caesar mushroom (*Amanita hemibapha*)	10	6	-
Hygroscopic earthstar mushroom (*Astraeus hygrometricus*)	4	2	-
Termite mushroom (*Termitomyces clypeatus* Heim)	10	5	-

^a^ Time of cooking (minute, min.). ^b^ Symbol “-” indicates not performed if the mushrooms are not eaten grilled (Food Consumption Data of Thailand [12]).

**Table 3 foods-12-02141-t003:** Parameters of mass spectrometer used for measuring vitamin D.

Form of Vitamin D	Parent Ion (*m*/*z*)	Daughter Ion (*m*/*z*)	Collision Energy (V)
Vitamin D2	397.5	379.38	9.42
Vitamin D3	385.5	367.39	11.02
Ergosterol	379.21	253.22	12.12
7-DHC	367.35	159.14	19.14
25-OH D2	395.35	377.38	14.9
25-OH D3	383.36	365.38	12.79
^2^H_3_D_2_ (internal standard)	400.4	382.46	9.89

**Table 4 foods-12-02141-t004:** Percentage of edible portion, yield factor, and moisture content of cultivated and wild mushrooms (*n* = 3) (Mean ± SD).

Mushroom	Type of Sample	Edible Portion (%)	Yield Factor	Moisture (g/100 g)
Cultivated mushrooms:				
Abalone mushrooms (*Pleurotus ostreatus* (FR.))	Raw	93 ± 1	- ^a^	92.17 ± 0.97
	Boiled	95 ± 1	0.97 ± 0.01	90.84 ± 0.88
	Stir-fried	93 ± 1	0.79 ± 0.01	84.18 ± 0.72
Bhutan oyster mushroom	Raw	88 ± 1	-	92.77 ± 0.20
(*Pleurotus sajor-caju* (Fr.) Singer)	Boiled	88 ± 1	0.72 ± 0.01	89.24 ± 0.71
	Stir-fried	80 ± 1	0.81 ± 0.01	84.47 ± 0.16
	Grilled	87 ± 1	0.61 ± 0.01	85.86 ± 2.54
Enokitake mushroom (*Flammulina velutipes*)	Raw	85 ± 2	-	90.44 ± 0.81
	Boiled	85 ± 4	0.75 ± 0.04	90.58 ± 0.77
	Stir-fried	89 ± 2	0.92 ± 0.03	83.68 ± 0.58
	Grilled	88 ± 1	0.88 ± 0.09	87.90 ± 1.73
Log black fungi mushroom	Raw	85 ± 3	-	90.65 ± 1.22
(*Lentinus squarrosulus* Mont)	Boiled	82 ± 2	0.85 ± 0.10	85.87 ± 0.58
	Stir-fried	84 ± 2	0.91 ± 0.02	82.79 ± 2.64
Log white fungi mushroom	Raw	100 ± 0	-	91.52 ± 1.48
(*Lentinus polychrous* Lev)	Boiled	100 ± 0	0.72 ± 0.03	87.99 ± 1.75
	Stir-fried	100 ± 0	0.94 ± 0.02	86.54 ± 2.00
Lung oyster mushroom (*Pleurotus pulmonarius*)	Raw	92 ± 2	-	91.49 ± 1.29
	Boiled	92 ± 2	0.76 ± 0.05	90.55 ± 1.49
	Stir-fried	93 ± 2	0.85 ± 0.04	84.94 ± 0.63
	Grilled	92 ± 2	0.67 ± 0.09	88.05 ± 1.31
Shiitake mushroom (*Lentinula edodes*)	Raw	98 ± 2	-	90.27 ± 1.29
	Boiled	96 ± 3	0.74 ± 0.05	87.59 ± 1.04
	Stir-fried	96 ± 3	0.89 ± 0.01	84.31 ± 1.41
	Grilled	95 ± 3	0.77 ± 0.01	86.90 ± 0.81
Straw mushroom (*Volvariella volvacea*)	Raw	96 ± 5	-	91.28 ± 1.82
	Boiled	95 ± 7	0.75 ± 0.03	89.77 ± 1.51
	Stir-fried	99 ± 1	0.86 ± 0.04	84.91 ± 0.42
	Grilled	99 ± 1	0.73 ± 0.08	88.29 ± 1.17
Wood ear mushroom (*Auricularia auricula-judae*)	Raw	98 ± 3	-	89.62 ± 2.20
	Boiled	98 ± 3	1.07 ± 0.0.04	91.77 ± 1.18
	Stir-fried	97 ± 5	0.93 ± 0.01	85.87 ± 1.57
Wild mushrooms:				
Half-dyed slender Caesar (*Amanita hemibapha*)	Raw	83 ± 7	-	96.04 ± 0.27
	Boiled	87 ± 2	0.61 ± 0.0.07	94.25 ± 0.32
	Stir-fried	84 ± 6	0.90 ± 0.02	89.58 ± 1.00
Hygroscopic earthstar (*Astraeus hygrometricus*)	Raw	100 ± 0	-	82.31 ± 1.47
	Boiled	100 ± 0	0.83 ± 0.0.02	79.97 ± 0.45
	Stir-fried	100 ± 0	0.92 ± 0.02	78.82 ± 0.01
Termite mushroom (*Termitomyces clypeatus* Heim)	Raw	92 ± 1	-	94.03 ± 0.24
	Boiled	92 ± 1	0.68 ± 0.0.04	94.71 ± 0.64
	Stir-fried	91 ± 1	0.80 ± 0.03	86.83 ± 1.04

^a^ Symbol “-” indicates as raw form (use original weight for calculation).

**Table 5 foods-12-02141-t005:** Ergosterol and vitamin D2 contents and true retentions of vitamin D2 of cultivated and wild mushrooms (*n* = 3) (Mean ± SD).

Mushroom	Type of Sample	Ergosterol (μg/100 g EP)	Vitamin D2 (μg/100 g EP)	True Retention of Vitamin D2 (%)
Cultivated mushrooms:				
Abalone mushrooms	Raw	12,170 ± 57	0.67 ± 0.02	-
(*Pleurotus ostreatus* (FR.))	Boiled	13,860 ± 82	0.73 ± 0.02	100.0 ± 0.0
	Stir-fried	16,200 ± 82	0.31 ± 0.02	37.8 ± 2.4
Bhutan oyster mushroom	Raw	8340 ± 163	0.07 ± 0.01	-
(*Pleurotus sajor-caju* (Fr.) Singer)	Boiled	15,053 ± 132	1.05 ± 0.11	100.0 ± 0.0
	Stir-fried	19,233 ± 176	1.22 ± 0.15	100.0 ± 0.0
	Grilled	22,347 ± 205	2.73 ± 0.16	100.0 ± 0.0
Enokitake mushroom	Raw	9943 ± 2159	0.06 ± 0.01	-
(*Flammulina velutipes*)	Boiled	11,037 ± 4317	0.06 ± 0.01	84.5 ± 14.0
	Stir-fried	13,140 ± 1923	0.07 ± 0.01	97.8 ± 3.9
	Grilled	13,687 ± 3209	0.06 ± 0.01	84.2 ± 13.6
Log black fungi mushroom (*Lentinus squarrosulus* Mont)	Raw	11,093 ± 1434	1.05 ± 1.25	-
	Boiled	14,433 ± 2655	0.43 ± 0.44	86.5 ± 17.2
	Stir-fried	13,503 ± 2176	0.38 ± 0.45	87.9 ± 13.2
Log white fungi mushroom (*Lentinus polychrous* Lev)	Raw	8480 ± 1819	2.31 ± 0.74	-
	Boiled	9823 ± 3757	0.47 ± 0.14	16.2 ± 9.2
	Stir-fried	11,140 ± 3020	0.24 ± 0.11	9.4 ± 1.9
Lung oyster mushroom (*Pleurotus pulmonarius*)	Raw	12,973 ± 545	15.88 ± 7.31	-
	Boiled	14,100 ± 516	12.35 ± 3.95	64.3 ± 30.9
	Stir-fried	18,877 ± 2112	4.98 ± 2.23	27.0 ± 5.6
	Grilled	17,870 ± 4050	11.29 ± 5.19	59.5 ± 36.0
Shiitake mushroom (*Lentinula edodes*)	Raw	12,070 ± 2969	0.17 ± 0.08	-
	Boiled	13,373 ± 4268	0.34 ± 0.27	85.9 ± 24.4
	Stir-fried	17,957 ± 3642	0.35 ± 0.34	89.5 ± 18.5
	Grilled	16,893 ± 2436	0.23 ± 0.13	74.5 ± 22.3
Straw mushroom (*Volvariella volvacea*)	Raw	11,637 ± 3820	0.94 ± 0.60	-
	Boiled	13,587 ± 4068	0.17 ± 0.15	4.3 ± 1.3
	Stir-fried	14,663 ± 4259	0.06 ± 0.01	4.4 ± 1.2
	Grilled	13,707 ± 4157	0.28 ± 0.28	5.1 ± 1.8
Wood ear mushroom (*Auricularia auricula-judae*)	Raw	10,223 ± 3113	1.99 ± 0.46	-
	Boiled	8937 ± 2650	1.62 ± 0.11	85.7 ± 17.9
	Stir-fried	11,397 ± 2037	1.00 ± 0.04	49.4 ± 14.1
Wild mushrooms:				
Half-dyed slender Caesar mushroom (*Amanita hemibapha*)	Raw	7713 ± 3619	0.39 ± 0.46	-
	Boiled	7340 ± 2313	0.36 ± 0.42	56.0 ± 7.5
	Stir-fried	9617 ± 1104	0.82 ± 0.57	94.7 ± 9.1
Hygroscopic earthstar mushroom (*Astraeus hygrometricus*)	Raw	17,273 ± 662	0.07 ± 0.01	-
	Boiled	19,947 ± 52	0.06 ± 0.01	73.2 ± 12.9
	Stir-fried	21,960 ± 684	0.33 ± 0.04	100.0 ± 0.0
Termite mushroom (*Termitomyces clypeatus* Heim)	Raw	13,044 ± 3004	7.15 ± 0.67	-
	Boiled	15,277 ± 1977	1.18 ± 0.78	11.4 ± 9.2
	Stir-fried	18,660 ± 3507	0.71 ± 0.76	7.6 ± 9.8

**Table 6 foods-12-02141-t006:** Estimated marginal means of vitamin D content and percentage of vitamin D true retention by the main effects of different species of mushrooms and cooking methods (calculated from two-way ANOVA) (*n* = 3).

Variables	Estimated Marginal Means ± Standard Error	
	Vitamin D (μg/100 g EP)	True Retention (%)
Different species of mushrooms:		
Abalone mushrooms (*Pleurotus ostreatus* (FR.))	0.06 ± 0.64 ^c,b^	68.9 ± 5.7 ^b,c^
Bhutan oyster mushroom (*Pleurotus sajor-caju* (Fr.) Singer)	1.27 ± 0.55 ^c,b^	100.0 ± 4.7 ^a^
Enokitake mushroom (*Flammulina velutipes*)	0.06 ± 0.55 ^c^	88.8 ± 4.7 ^a,b^
Log black fungi mushroom (*Lentinus squarrosulus* Mont)	0.33 ± 0.64 ^c,b^ 33	87.2 ± 5.7 ^a,b^
Log white fungi mushroom (*Lentinus polychrous* Lev)	1.01 ± 0.64 ^c,b^ 33	12.8 ± 5.7 ^d^
Lung oyster mushroom (*Pleurotus pulmonarius*)	11.45 ± 0.55 ^a^	50.3 ± 4.7 ^c^
Shiitake mushroom (*Lentinula edodes*)	0.27 ± 0.55 ^c,b^	83.3 ± 4.7 ^a,b^
Straw mushroom (*Volvariella volvacea*)	0.36 ± 0.55 ^c,b^	4.6 ± 4.7 ^d^
Wood ear mushroom (*Auricularia auricula-judae*)	1.53 ± 0.64 ^c,b^	67.6 ± 5.7 ^b,c^
Half-dyed slender Caesar mushroom (*Amanita hemibapha*)	0.52 ± 0.64 ^c,b^	75.4 ± 5.7 ^a,b,c^
Hygroscopic earthstar mushroom (*Astraeus hygrometricus*)	0.15 ± 0.64 ^c,b^	86.6 ± 5.7 ^a,b^
Termite mushroom (*Termitomyces clypeatus* Heim)	3.01 ± 0.64 ^b^	9.5 ± 5.7 ^d^
Cooking methods in different species of mushrooms:		
Raw	2.51 ± 0.32 ^a^	-
Boiling	1.56 ± 0.32 ^a,b^	64.0 ± 2.3 ^a^
Stir-frying	0.87 ± 0.32 ^b^	58.8 ± 2.3 ^a^
Grilling	3.13 ± 0.49 ^a^	64.7 ± 3.6 ^a^

Values with unlike superscript letters of species of mushrooms or cooking methods in the same column were significantly different for a given variable (*p* < 0.05 two-way ANOVA followed by Tukey’s HSD post hoc multiple comparisons). The marginal averages of the ergosterol content are not presented.

## Data Availability

Data are contained within the article.

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
