# Peer review of "Vitamin D Content in Commonly Consumed Mushrooms in Thailand and Its True Retention after Household Cooking"

_foods, 2023, doi:10.3390/foods12112141_

Round 1

Reviewer 1 Report

In general, the topic of the manuscript is interesting since there are a lot of studies on vitamin D3 sources and stability, while information on vitamin D2 sources and stability is lacking.

Table 1 illustrates the relationships among English name, Thai name and scientific name of mushrooms, which is useful. On the other hand, for the following tables and Figures I suggest to use the scientific names in order to facilitate the readers.

Table 5 2I suggest to comment the content of vitamin D2 in raw and processed mushroom species with reference to vitamin D fortified foods (see: Lavelli, V.; D’Incecco, P.; Pellegrino, L. Vitamin D incorporation in foods: Formulation strategies, stability, and bioaccessibility as affected by the food matrix. Foods 2021, 10, 1989). Interestingly, it seems that some mushroom species possess a natural content of vitamin D which is in the range of that proposed for functional foods.

 Line 346

I suggest to change “cooking” to “cooking procedures”

Author Response

Comments and Suggestions for Authors

Response to Comments and Suggestions

Reviewer 1: In general, the topic of the manuscript is interesting since there are a lot of studies on vitamin D3 sources and stability, while information on vitamin D2 sources and stability is lacking.

Table 1. Illustrates the relationships among English name, Thai name and scientific name of mushrooms, which is useful. On the other hand, for the following tables and Figures I suggest to use the scientific names in order to facilitate the readers.

All tables and figures were modified by adding the scientific name.

Table 5. I suggest to comment the content of vitamin D2 in raw and processed mushroom species with reference to vitamin D fortified foods (see: Lavelli, V.; D’Incecco, P.; Pellegrino, L. Vitamin D incorporation in foods: Formulation strategies, stability, and bioaccessibility as affected by the food matrix. Foods 2021, 10, 1989). Interestingly, it seems that some mushroom species possess a natural content of vitamin D which is in the range of that proposed for functional foods.

More discussion and information was added in the revised manuscript as detailed below. “Lung oyster mushrooms and termite mushrooms contained high levels of vitamin D2 (15.88 ± 7.31 and 7.15 ± 0.67 µg per 100 g EP respectively).  Lavelli et al., 2021 summarized in vivo studies on the effects of vitamin-D-fortified foods. There is a limit study using vitamin D2 for in vivo study. Vitamin D2 from UV-irradiated yeast in bread (25 µg/d) did not raise serum 25(OH)D concentration. Biancuzzo et al., (2012) indicated the equally bioavailable of vitamin D2 and vitamin D3 in Ca++-enriched orange juice and capsules. Vitamin D3 supplements doses were shown an effect in in-creasing serum 25(OH)D concentration for example 10 µg/d in Ca++ enriched yogurt studied in healthy women aged >65 years, 10.8 µg/d in low-fiber wheat bread and 12.3 µg/d in high-fiber sourdough rye bread studied in healthy women aged 25-45 years [ ], 15 µg/d in cheese healthy women aged 25–45 years. So, natural vitamin D in these mushrooms are in the range of the fortified functional foods.” 

Line 346.  I suggest to change “cooking” to “cooking procedures”

The title of section 3.6 was modified as suggestion into “Effect of different cooking procedure on vitamin D in mushrooms”

Reviewer 2 Report

The authors presented an experimental work on wild and cultivated mushrooms available in Thailand to evaluate the concentrations of ergosterol and ergocalciferol (Vit D2).

From the analyzes carried out using a high-resolution detection system (LC-MS/MS), the authors highlighted how the concentrations varied according to the cooking method, showing a fairly homogeneous vitamin D content among the types of mushrooms studied, except two species (one cultivated and one wild).

The manuscript is well structured and the topic is interesting. References are adequate. I have a few observations which I list below:

- The title should specify that it refers to mushrooms available in Thailand.

- As vitamin D concentrations vary according to the part of the mycelium, it should be specified which parts were used in the study

- It is unclear whether the results are from single samples or if replicates were made for testing and how much. Sampling is a particularly delicate aspect that could significantly affect the results.

- In the manuscript and the abstract it would be appropriate in the values presented that the comma was always used to separate the thousands

- In line 311 it would be more correct to refer to non-animal foods and not to mycelium

- Table 5 shows a drastic increase in the concentration of vitamin d2 and ergosterol in Bhutan oyster mushrooms. Since increasing the concentrations is unreasonable, there is weight loss derived from cooking. It would be better, in the case of concentrations after cooking, to normalize for the raw weight and not for the cooked weight of which it is not known how much the reduction in mass has been. In this way, the real vitamin D retention could be better understood. In any case, this increase in this circumstance should be discussed

- Since there does not seem to be a homogeneous trend of higher vitamin D concentrations in wild species, and considering that the cultivation conditions should not be favourable for biosynthesis, the authors should propose a mechanism explaining this unexpected finding.

- In the captions of Table 6, reference is also made to the marginal averages of the ergosterol content which, however, is not present in the table

Author Response

Comments and Suggestions for Authors

Response to Comments and Suggestions

Reviewer 2:

The authors presented an experimental work on wild and cultivated mushrooms available in Thailand to evaluate the concentrations of ergosterol and ergocalciferol (Vit D2).

From the analyzes carried out using a high-resolution detection system (LC-MS/MS), the authors highlighted how the concentrations varied according to the cooking method, showing a fairly homogeneous vitamin D content among the types of mushrooms studied, except two species (one cultivated and one wild).

The manuscript is well structured and the topic is interesting. References are adequate. I have a few observations which I list below:

- The title should specify that it refers to mushrooms available in Thailand.

The name of manuscript was modified as suggestion into “Vitamin D content in commonly consumed mushrooms in Thailand and their stability after household cooking “.

- As vitamin D concentrations vary according to the part of the mycelium, it should be specified which parts were used in the study.

Edible part is the combined 3 parts including stalk, cap, and gills. This is mentioned in section 3.2 as…”most of the edible portion is comprised of the partial veil including part of the stalk, cap, and gills”.

- It is unclear whether the results are from single samples or if replicates were made for testing and how much. Sampling is a particularly delicate aspect that could significantly affect the results.

- Vitamin D were analyzed in duplicate from 3 markets as mentioned in section 2.5. Quality control system of vitamin D analysis. “Each mushroom sample was analyzed in duplicate along with the QC sample (SRM).”

- It is also added in the sentence of the first paragraph of section 2.5 as “All samples were analyzed, in duplicate, at the accredited ISO/IEC 17025:2017 laboratory of the Institute of Nutrition, Mahidol University.”

- I agreed to the reviewer’s comment. So, this study selected 3 representative markets to investigate this variation as mentioned in 2.2. Selection and collection of cultivated and wild mushrooms. “The cultivated species were purchased from three main markets and three sellers (n=3) covering the Thai-Market (representative of east and/or northern parts of Thailand), Klong-toey market (representative of the Bangkok area), and Railway Bangkok-Noi Market or Sala-Num-Ron market (representative of southern parts of Thailand).”

- In the manuscript and the abstract it would be appropriate in the values presented that the comma was always used to separate the thousands.

All number with thousand was modified by adding “,” to separate the thousands in abstract, Table 5, and throughout manuscript.

- In line 311 it would be more correct to refer to non-animal foods and not to mycelium.

The word of “mycelium” is changed to non-animal foods as “Mushrooms are the only non-animal foods that can make vitamin D.”. 

- Table 5 shows a drastic increase in the concentration of vitamin D2 and ergosterol in Bhutan oyster mushrooms. Since increasing the concentrations is unreasonable, there is weight loss derived from cooking. It would be better, in the case of concentrations after cooking, to normalize for the raw weight and not for the cooked weight of which it is not known how much the reduction in mass has been. In this way, the real vitamin D retention could be better understood. In any case, this increase in this circumstance should be discussed

- The true retention (section 2.6) is used and calculated as a true retention which weight before and after cooking were used for this calculation.  Another term as apparent retention may be presented which the weight change after cooking were not involved.  However, true retention is recommended by INFOOD, FAO.

More discussion was added in the revised manuscript.

- For Bhutan oyster mushrooms, vitamin D2 was remarkably increased after boiling, stir-frying, and grilling (15, 17, and 39 times compared to the raw, respectively) whereas ergosterol increased only 2-3 times.  This may be when certain foods that contain vitamin D are cooked, the heat can cause some of the vitamin D that is present to become more available and easier to extract and detect. Another reason is when mushrooms are exposed to heat, they may convert a form of vitamin D called ergosterol into a more usable form of vitamin D2.

- Since there does not seem to be a homogeneous trend of higher vitamin D concentrations in wild species, and considering that the cultivation conditions should not be favourable for biosynthesis, the authors should propose a mechanism explaining this unexpected finding.

More discussion of termite mushrooms was added in the section 3.6. (Vitamin D content) of the revised manuscript. For example,

- Vitamin D2 in termite mushrooms (wild mushrooms) was found a high variation of ergosterol and vitamin D2 (13,044 ± 3,004 and 7.15 ± 0.67 µg/100 g EP, about 23% and 9% RSD respectively). This mushroom was an underground mushroom and typically lo-cated near the base of trees and termite areas. It was collected in the different areas of the conservatively studied area. The intensity of sunlight and UV radiation that the mushrooms are exposed to during growth can impact the high variation of production of vitamin D. The levels of vitamin D in termite mushrooms can also vary depending on the season, it was collected during the summer season.  Overall, the levels of vita-min D in termite mushrooms can vary due to various factors, including geographic lo-cation, growing conditions, and seasonal variability.

- In the captions of Table 6, reference is also made to the marginal averages of the ergosterol content which, however, is not present in the table.

Title of Table 6 was modified by remove the word “ergosterol” from the title and explanation under the table.

Reviewer 3 Report

Well written article, some suggestions

1. Discussion must be improved to compare your data with previous work.

2. Why these mushrooms were chosen instead of others?

3.  Provide graphical abstract

4. Any pictures of cooking method and what was the best cooking parameters to get the best Vitamin D?

5. What do you mean by stability of Vitamin D? how do you calculate stability?

Author Response

Comments and Suggestions for Authors

Response to Comments and Suggestions

Reviewer 3:

Well written article, some suggestions

1. Discussion must be improved to compare your data with previous work.

Discussion was added in the revised manuscript as response in reviewer 1 and 2.

2. Why these mushrooms were chosen instead of others?

Criteria for choosing mushrooms was mentioned in section 2.2. (Selection and collection of cultivated and wild mushrooms).  More information are also added as “Nine species of the most commonly consumed cultivated mushrooms and three species of wild mushrooms (Figure 1) were selected based on combined data from the most commonly consumed of Food consumption data of Thailand (2016) [12] and lack of database in Food composition database of Thailand [13].”

3.  Provide graphical abstract

Graphical abstract is created and added in the system.

4. Any pictures of cooking method and what was the best cooking parameters to get the best Vitamin D?

As mention in conclusion as “True retention levels of vitamin D2 in boiled, stir-fried, and grilled mushrooms were not significantly different.  Cooking methods may cause a significant loss of vitamin D2, but the degree of loss depends on the physical characteristics of a mushroom, the heating process, and cooking time.  Cooked lung oyster mushroom is the most suitable for promotion as a high source of vitamin D, since it is easy to consume, has a delicious taste, and is inexpensive. It should be promoted especially for persons who have limited exposure to the sun and thus require more vitamin D-rich foods.“  Therefore, it cannot select the best cooking for retain vitamin A.

5. What do you mean by stability of Vitamin D? how do you calculate stability?

The “stability” in this title is emphasized on the change of vitamin D after cooking. Therefore, “true retention” was replaced of “stability”.

Original title “Vitamin D content in commonly consumed mushrooms and their stability after household cooking” was modified as “Vitamin D content in commonly consumed mushrooms in Thailand and their true retention after household cooking”.

Round 2

Reviewer 1 Report

The points raised by this reviewer have been addressed

Author Response

According to your comments, the points raised by this reviewer have been addressed. Thank you very much.  Could you please accept this manuscript for further processing?

Reviewer 3 Report

I have agree on most of the responses especially on cooking aspects. However, i noticed some slight drawbacks that need minor improvements.

Introduction: no relation of Mushroom Vitamin D with SDG, perhaps zero hunger ?, consider for improvements https://doi.org/10.1016/j.fshw.2022.07.040

Discussion: No direct citations of Thailand mushrooms with Asian countries, consider for improvements.

Minor checks

Author Response

  1. Comment on Introduction: no relation of Mushroom Vitamin D with SDG, perhaps zero hunger ?, consider for improvements https://doi.org/10.1016/j.fshw.2022.07.040

Response: More information was added in the introduction as “Edible mushrooms are a good source of vitamin D and other nutrients which can be linked to Sustainable Development Goals (SDGs) number 3 good health and well-being. In addition, mushrooms are also involved in SDG 2 Zero Hunger due to mushrooms are easy to cultivate, promote consumption, and provide an opportunity to diversify diets, particularly for communities with limited access to animal-based proteins.”.

  1. Comment on Discussion: No direct citations of Thailand mushrooms with Asian countries, consider for improvements.

Response: A reference of study mushrooms in Thailand was added. More discussion is added as “Banlangsawan & Sanoamuang (2016) studied the effect of UV-B irradiation in Thai edible mushrooms namely enokitake mushroom (Flammulina velutipes), log black fungi mush-room (Lentinus polychrous Lev), log white fungi mush-room (Lentinus squarrosulus Mont), wood ear mushroom (Auricularia auricula-judae), Abalone mushroom (Pleurotus ostreatus (FR.)), Lung Oyster mushroom (Pleurotus pulmonarius) which these raw mushrooms found vitamin D2 less than 4 μg/g of dry weight.”.
